# Trends in harmful drug exposure during pregnancy in France between 2013 and 2019: A nationwide cohort study

Margaux Louchet[1,2]*, Mathis Collier[1,3◉], Nathanaël Beeker[1,3◉], Laurent Mandelbrot[4,5], Jeanne Sibiude[4,5‡], Laurent Chouchana[1,6‡], Jean Marc Treluyer[1,3]

1 UPR7323 "Pharmacology and Drug Evaluatioán in Children and Pregnant Women", Université Paris Cité, Paris, Île-de-France, France, 2 Fédération Hospitalo-Universitaire PREMA, Université Paris Cité, Paris, Île-de-France, France, 3 Clinical Research Unit, Université de Paris CIC P1419, Assistance Publique-Hôpitaux de Paris, Paris, Île-de-France, France, 4 Department of Obstetrics and Gynecology, Louis Mourier Hospital, Assistance Publique-Hôpitaux de Paris, Paris, Île-de-France, France, 5 INSERM Infection, Antimicrobials, Modelling, Evolution U1137, Université Paris Cité, Paris, Île-de-France, France, 6 Department of Perinatal Pediatric and Adult Pharmacology, Regional Pharmacovigilance Center, Cochin Hospital, Assistance Publique-Hôpitaux de Paris, Paris, Île-de-France, France

◉ These authors contributed equally to this work.
‡ JS and LC also contributed equally to this work.
* margaux.louchet@gmail.com

**Data Availability Statement:** All relevant data are within the paper and its Supporting Information files.

## Abstract

### Objective

Describe the trends of exposure to harmful drugs during pregnancy over recent years in France.

### Design

Nationwide cohort study.

### Setting

The French National administrative health Data System (SNDS).

### Population

Pregnancies starting between 2013 and 2019 and outcomes corresponding to live births, medical terminations of pregnancy, and stillbirths.

### Methods

Each pregnancy was divided into a preconceptional period of 90 days before conception and three trimesters from conception to birth. Harmful drugs were defined according to their risks to the fetus: teratogenicity or fetotoxicity. Exposure was defined using the critical period during pregnancy for each type of harmful drug: preconceptional period or first trimester for teratogenic drugs and second or third trimesters for fetotoxic drugs.

**Funding:** The authors received no specific funding for this work.

**Competing interests:** The authors have declared that no competing interests exist.

## Main outcome measures

Prevalence of pregnancies exposed to at least one harmful drug.

## Results

Among 5,253,284 pregnancies, 204,402 (389 per 10,000) pregnancies were exposed to at least one harmful drug during the critical periods: 48,326 (92 per 10,000) pregnancies were exposed to teratogenic drugs during the preconceptional period or the first trimester, and 155,514 (299 per 10,000) pregnancies were exposed to fetotoxic drugs during the second or third trimesters. Teratogenic drugs were mainly retinoids for topical use (44 per 10,000 pregnancies), antiepileptics (13 per 10,000 pregnancies) and statins (13 per 10,000 pregnancies). Fetotoxic drugs were mainly non-steroidal anti-inflammatory drugs (NSAIDs), for systemic (128 per 10,000 pregnancies) and topical use (122 per 10,000 pregnancies). Exposure to teratogenic drugs decreased from the preconceptional period to the first trimester. Exposure to fetotoxic drugs decreased from the second to the third trimester. Between 2013 and 2019, we found a decrease in harmful drug exposure overall, mainly for topical and systemic NSAIDs and for topical retinoids.

## Conclusions

In this nationwide study, about one in 25 pregnancies was exposed to at least one harmful drug, mainly NSAIDs and topical retinoids. Although the prevalence of harmful drug exposure decreased over the study period, NSAID exposure in the second and third trimester remains of concern.

## Introduction

Drug exposure is common during pregnancy, although data regarding safety for women and offspring are limited [1]. As illustrated by the thalidomide tragedy in the previous century, drugs may be harmful to the developing fetus and result in major consequences such as congenital malformations (phocomelia in the case of the thalidomide embryopathy) [2]. Moreover, drugs can cause functional defects such as illustrated by the neurodevelopmental adverse effects of valproate [3–5] or renal failure caused by non-steroidal anti-inflammatory drugs (NSAIDs) [6–9]. Hence harmful drugs can be separated in two categories according to their risk profiles and depending on the critical period of exposure during pregnancy: (i) Teratogenic drugs cause congenital malformations if exposure occurs during embryogenesis, i.e. in the first trimester; (ii) Fetotoxic drugs cause functional alterations in organ or tissue functions when taken during the second or third trimester [10, 11]. Teratogenic drug exposure stopped shortly before conception is also of concern given the possible persistence of the drug in the body during early embryogenesis, perhaps before the mother becomes aware she is pregnant.

In high income countries, exposure to prescription drugs during pregnancy ranges from 57% to 99% (including vitamins and minerals) with variations among countries, and seems higher in France than in other European countries [12, 13]. Hence, recent studies reported that, in France, 70% to 99% of pregnant women were exposed to at least one drug, with an average of 9 to 11 different drugs per pregnancy [1, 14]. The prevalence of pregnant women filling a prescription for a potentially harmful drug differed widely according to the type of

data source (e.g.: claim databases or questionnaires), or risk classifications. Overall, most studies of harmful drug exposure during pregnancy showed a higher use in early pregnancy compared to later trimesters [15–17].

In countries members of the Organization for Economic Cooperation and Development (OECD), prescription prevalence of drugs with positive evidence of human fetal risks for which risks outweigh potential benefits (category X of the former FDA classification) ranged from 1.0% to 4.9% [12]. In the US during the 1996–2000 period, 5.8% of the pregnancies were exposed to a drug for which potential benefits might warrant use of the drug (category D of the former FDA classification, including NSAIDs during the 3$^{rd}$ trimester) [15], while it was 6.3% in Quebec during the 1998–2002 period [18], and 2.0% in Australia during the 2005–2012 period [19]. A French nationwide cohort study between 2016 and 2017, assessed that exposure to potentially harmful drugs including NSAIDs, concerned 2.2% or 3.9% of pregnancies, according to the Swedish or Australian risk classification systems, respectively [17]. The authors reported that NSAIDs was the most prescribed harmful drug class. Another French cohort study on a random sample of the general population estimated that 2.3% of the pregnancies were exposed to NSAIDs after the sixth month of gestation, during the critical period when these drugs induce renal insufficiency and closure of the ductus arteriosus and are therefore strictly contraindicated [16].

In this nationwide cohort study, we assessed exposure to harmful drugs in pregnant women during the respective critical periods: teratogenic drugs before conception or during the first trimester of pregnancy, and fetotoxic drugs during the second or third trimester of pregnancy. We also assessed trends of exposure over the recent years.

## Methods

### Data source

This study was conducted using the French national administrative health data system (*Système National des Données de Santé*, SNDS) which covers 98.9% of the French population [20]. It is a fully anonymized database based on the national health care reimbursement database linked to the national hospital discharge database [21]. It comprises exhaustive data on patient health care expenditure such as dispensed drugs or outpatient medical care, and chronic diseases. It also comprises a proxy for economic status, the affiliation to CMUc, which is a special national insurance allowing free access to health care for people with low income. Drugs are encoded using the Anatomic-Therapeutic-Chemical classification (ATC) and the French drug identification number code CIP. Moreover, it comprises diagnosis of all hospital admissions in France, classified according to the International Classification of Diseases, 10th Revision (*ICD-10; www.who.int/classifications/icd/en/*).

Pregnancy outcomes such as live births, medical terminations of pregnancy or stillbirths are encoded as discharge diagnoses and as medical procedures with the CCAM (French National health insurance's common classification of medical acts).

### Study population

We included all pregnancies with a date of conception between January 2013 and December 2019, resulting in live births, terminations of pregnancy for fetal or maternal medical reasons, or stillbirths. We did not include ectopic nor molar pregnancies (S1 Table). We excluded (i) fetal losses before 22 weeks of gestation because of the lack of representativeness regarding these situations in the SNDS and (ii) pregnancies in which the maternal age was above 55 years.

The date of conception was calculated using 2 variables encoded in the hospital discharge database at birth: the date of pregnancy outcomes and the numbers of days since the last menstrual period. In the infrequent cases of missing data for this last variable, the date of conception was estimated using the date of pregnancy outcomes and the gestational age at birth according to the physician's assessment at birth.

Each pregnancy was divided into three periods: the first trimester (T1) from conception until day 91, the second trimester (T2) between day 92 and day 182, and the third trimester (T3) between day 183 and delivery. Furthermore, a preconceptional period (PC) was defined as the 90-day period before the date of conception (day 0).

Characteristics of pregnant women (age at pregnancy outcome, comorbidities, CMUc beneficiaries) and related to the pregnancy (type of pregnancy outcomes and gestational age at birth) were described. Comorbidities such as pre-gestational diabetes, hypertension, and psychiatric history were identified using discharge codes or reimbursed filled prescription corresponding to each disease in the year prior or during pregnancy (S1 Table).

## Drug exposure

Harmful drugs were defined based on the French summary of product characteristics (SmPC) and on our clinical or pharmacological knowledge. Depending on the type of drug, harmfulness concerned either a whole pharmacological class (e.g., NSAIDs) or only specific molecules among a pharmacological class (e.g., valproic acid among antiepileptic drugs).

They were further classified in two groups, according to the type of harm to the fetus: teratogenic or fetotoxic drugs (S2 Table). Exposure was described considering the related critical periods: preconceptional period or T1 for teratogenic drugs and T2 or T3 for fetotoxic drugs. Exposure over a critical period was defined as filling at least one reimbursement of one drug during the period of interest. It was expressed as prevalence (per 10,000 pregnancies), i.e., the number of pregnancies with at least one reimbursement divided by the total number of pregnancies during the critical periods of interest. Years of exposure referred to years of conception for each pregnancy. Treatment dose was not considered in this study.

## Ethics statement

The SNDS is a fully anonymised ongoing database, containing information on all claims reimbursed by the French National Health Insurance. Patients are affiliated from birth to death with a unique patient identifier. Our study cohort was approved by the national personal data protection agency *Commission Nationale de l'Informatique et des Libertés* (CNIL) (agreement DE-2015-192) and the French committee for the protection of health care data named *Comité spécifique de Recherche sur les Données de Santé* (CERESS). In agreement with French regulations, observational studies conducted on anonymous medico-administrative data did not require an ethics committee approval.

## Statistical analysis

In this study, the unit of analysis was the pregnancy (one woman could account for one or more pregnancies). Sensitivity analyses were performed considering only harmful drugs administered orally. Population characteristics were presented as mean +/- standard deviation (SD) and with percentages. Trends of drug exposures through the study period was assessed with $\chi^2$ test. P values of $< 0.05$ were deemed significant. All analyses were performed with R Studio software version 4.1.2.

## Results

### Maternal characteristics and pregnancy outcomes

A total of 5,253,284 pregnancies were included in the cohort from 2013 to 2019 (S1 Fig). In the overall population, 5,129,561 (97.6%) pregnancies resulted in live births, 97,345 (1.8%) pregnancies in medical terminations of pregnancy and 26,378 (0.5%) pregnancies in stillbirths (Table 1, S3 Table). Mean maternal age at delivery was 30.3 +/- 5.4 years. We found a history of psychiatric disorders in 4.0% of pregnancies, hypertension in 1.4%, and pre-gestational diabetes in 0.7%. We found low-income status in 4.6% of pregnancies. For live births, mean gestational age was 39.0 +/- 1.9 weeks of gestation, with 6.2% preterm births (before 37 weeks' gestation).

**Table 1. Maternal characteristics and pregnancy outcomes according to exposure to harmful drugs.**

|  | Overall n = 5,253,284 | Exposed pregnancies to harmful drugs n = 204,402 | Unexposed pregnancies to harmful drugs n = 5,048,882 |
|---|---|---|---|
| Prevalence over whole cohort | - | 3.9% | 96.1% |
| Pregnant women | 4,074,996 | 197,356 | 3,952,839 |
| **Maternal age (years)** |  |  |  |
| Mean (+/- SD) | 30.3 +/- 5.4 | 30.3 +/- 5.8 | 30.2 +/- 5.4 |
| < 20 | 110,055 (2.1%) | 5,607 (2.7%) | 104,448 (2.1%) |
| 20–29 | 2,247,780 (42.8%) | 86,350 (42.6%) | 2,161,430 (42.8%) |
| 30–39 | 2,659,845 (50.6%) | 100,173 (49.0%) | 2,559,672 (50.7%) |
| ≥ 40 | 235,604 (4.5%) | 12,272 (6.0%) | 223,332 (4.4%) |
| **Maternal comorbidities** |  |  |  |
| Psychiatric troubles | 212,278 (4.0%) | 16,204 (7.9%) | 196,074 (3.9%) |
| Pre-gestational diabetes | 36,270 (0.7%) | 3,568 (1.8%) | 32,702 (0.7%) |
| Hypertension | 74,276 (1.4%) | 8,076 (4.0%) | 66,200 (1.3%) |
| **Number of hospitalisations in the year prior to pregnancy** |  |  |  |
| Mean (+/- SD) | 0.3 +/- 1.2 | 0.5 +/- 1.3 | 0.3 +/- 1.2 |
| none | 4,033,101 (76.8%) | 145,173 (71.0%) | 3,887,928 (77.0%) |
| 1 | 883,643 (16.8%) | 40,584 (19.9%) | 843,059 (16.7%) |
| 2 or more | 336,540 (6.4%) | 18,645 (9.1%) | 317,895 (6.3%) |
| Low-income status* | 240,564 (4.6%) | 15,905 (7.8%) | 224,659 (4.5%) |
| **Pregnancy outcomes** |  |  |  |
| Live births | 5,129,561 (97.6%) | 200,009 (97.9%) | 4,929,552 (97.6%) |
| Medical termination <22G | 79,071 (1.5%) | 2,465 (1.2%) | 76,606 (1.5%) |
| Medical termination≥22GW | 18,274 (0.4%) | 733 (0.4%) | 17,541 (0.4%) |
| Stillbirths | 26,378 (0.5%) | 1,195 (0.6%) | 25,183 (0.5%) |
| **Gestational age at birth (for live births only)** |  |  |  |
| Mean (+/- SD) | 39.0 +/- 1.9 | 39.0 +/- 1.9 | 39.0 +/- 1.9 |
| Premature birth < 37GW | 323,042 (6.2%) | 13,288 (6.5%) | 309,754 (6.1%) |
| <28GW | 15,848 (0.3%) | 589 (0.3%) | 15,259 (0.3%) |
| [28–31] GW | 30,236 (0.6%) | 1,216 (0.6%) | 29,020 (0.6%) |
| [32–36] GW | 276,958 (5.3%) | 11,483 (5.6%) | 265,475 (5.3%) |

*Low-income status was defined as affiliation to CMUc

**Abbreviations:** CMU (*couverture maladie universelle*), GW (gestational week), SD (standard deviation)

Data are shown as mean (+/- SD) or n (%)

## Harmful drug exposure

During the study period, 204,402 (389 per 10,000) pregnancies were exposed to at least one harmful drug between the preconceptional period and the delivery.

## Teratogenic drugs

Regarding teratogenic drugs, 48,326 (92 per 10,000) pregnancies were exposed during the preconceptional period or the first trimester (Table 2). The most important exposure for teratogenic drugs were retinoids for topical use (44 per 10,000), teratogenic antiepileptic drugs (AEDs) (13 per 10,000) and HMG Co-A reductase inhibitors (statins) (13 per 10,000) (S4 Table). We observed a decrease in the number of exposed pregnancies from 40,495 (77 per 10,000) in the preconceptional period to 17,773 (34 per 10,000) in the first trimester. The

**Table 2. Maternal characteristics and pregnancy outcomes among pregnancies exposed to a teratogenic drug.**

|  | During preconceptional period or T1 | During preconceptional period | During T1 |
|---|---|---|---|
| **Exposed pregnancies to a teratogenic drug, n (% over all pregnancies)*** | 48,326 (0.9%) | 40,495 (0.8%) | 17,773 (0.3%) |
| **Exposed pregnant women** | 46,628 | 39,117 | 17,153 |
| **Maternal age (years)** |  |  |  |
| **Mean (+/- SD)** | 30.9 +/- 5.5 | 30.9 +/- 5.5 | 31.4 +/- 5.8 |
| < 20 | 688 (1.4%) | 515 (1.3%) | 279 (1.6%) |
| 20–29 | 19,490 (40.3%) | 16,161 (39.9%) | 6,490 (36.5%) |
| 30–39 | 25,005 (51.7%) | 21,151 (52.2%) | 9,494 (53.4%) |
| ≥ 40 | 3,143 (6.5%) | 2,668 (6.6%) | 1,510 (8.5%) |
| **Maternal comorbidities** |  |  |  |
| **Psychiatric troubles** | 7,002 (14.5%) | 6,293 (15.5%) | 3,191 (18.0%) |
| **Pre-gestational diabetes** | 1,764 (3.7%) | 1,577 (3.9%) | 831 (4.7%) |
| **Hypertension** | 3,886 (8.0%) | 3,535 (8.7%) | 1,776 (10.0%) |
| **Number of hospitalisations in the year prior to pregnancy** |  |  |  |
| **Mean (+/- SD)** | 0.6 +/- 1.9 | 0.6 +/- 1.8 | 0.6 +/- 2.3 |
| none | 33,233 (68.8%) | 27,579 (68.1%) | 12,020 (67.6%) |
| 1 | 9,690 (20.1%) | 8,224 (20.3%) | 3,616 (20.4%) |
| 2 or more | 5,403 (11.2%) | 4,692 (11.6%) | 2,137 (12.0%) |
| **Low-income status**** | 2,812 (5.8%) | 2,276 (5.6%) | 1,108 (6.23%) |
| **Pregnancy outcome** |  |  |  |
| **Live births** | 46,486 (96.2%) | 38,910 (96.1%) | 16,945 (95.3%) |
| **Medical termination <22GW** | 1,262 (2.6%) | 1,086 (2.7%) | 568 (3.2%) |
| **Medical termination≥22GW** | 231 (0.5%) | 198 (0.5%) | 114 (0.6%) |
| **Still births** | 347 (0.7%) | 301 (0.7%) | 146 (0.8%) |
| **Gestational age at birth (for live births only)** |  |  |  |
| **Mean (+/- SD)** | 38.8 +/- 2.1 | 38.8 +/- 2.1 | 38.7 +/- 2.2 |
| **Premature birth < 37GW** | 4,133 (8.6%) | 3,528 (8.7%) | 1,672 (9.4%) |
| <28GW | 224 (0.5%) | 195 (0.5%) | 101 (0.6%) |
| [28–31] GW | 466 (1.0%) | 409 (1.0%) | 190 (1.1%) |
| [32–36] GW | 3,443 (7.1%) | 2,924 (7.2%) | 1,381 (7.8%) |

* Whole cohort study includes n = 5,253,284 pregnancies at preconceptional period and 1st trimester

** Low-income status was defined as affiliation to CMUc

**Abbreviations:** CMU (*couverture maladie universelle*), GW (gestational week), SD (standard deviation)

Data are shown as mean (+/- SD) or n (%)

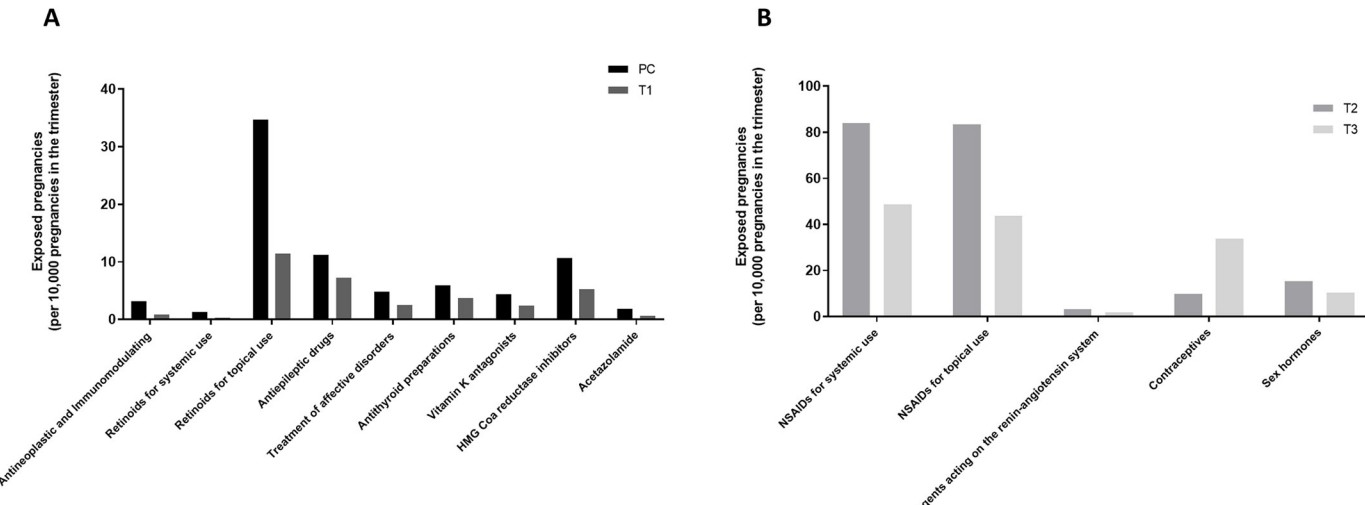

**Fig 1. Exposure to harmful drugs during pregnancy in France between 2013 and 2019. A. Exposure to teratogenic drugs during the preconceptional period (PC) or the first trimester (T1).** Antineoplastic and Immunomodulating drugs: fingolimod, leflunomide, lenalidomide, methotrexate, mycophenolic acid, thalidomide, teriflunomide. Retinoids for systemic use: acitretin (for psoriasis treatment), alitretinoin agents for dermatitis, etretinate, isotretinoin for systemic use, tretinoin. Retinoids for topical use: adapalene, alitretinoin antineoplastic agent, isotretinoin, tifarotene, tretinoin. Antiepileptic drugs: carbamazepine, fosphenytoin, oxcarbazepine, phenytoin, topiramate, valproic acid. Treatment of affective disorders: divalproate, lithium, valpromide. Antithyroid preparations: carbimazole, thiamazole. Vitamin K antagonists: acenocoumarol, fluindione, warfarin. HMG Coa reductase inhibitors: atorvastatin, fluvastatin, lovastatin, pitavastatin, pravastatin, rosuvastatin, simvastatin. Other drugs acting as teratogens: acetazolamide, antiglaucoma preparation. **B. Exposure to fetotoxic drugs during the second (T2) or the third trimester (T3).** Non-steroids anti-inflammatory drugs for systemic use: acetic acid derivatives and related substances, oxicams, propionic acid derivatives, fenamates, coxibs, other anti-inflammatory and antirheumatic agents, non-steroids, acetylsalicylic acid $\geq$ 250mg. Non-steroids anti-inflammatory drugs for topical use. Agents acting on the renin-angiotensin system: Angiotensin Converting Enzyme inhibitors, angiotensin II receptor blockers, renin inhibitors. Contraceptives: hormonal contraceptives for systemic use, emergency contraceptives, contraceptives for topical use. Sex hormones: androgens, estrogens, progestogens, anti-androgen, hormone replacement therapy, gonadotropins and other ovulation stimulants, antigonadotropins and similar agents, sex hormones for systemic disease. For details regarding specific drugs, refer to Supplementary, S2 Table.

decrease from the preconceptional period to the first trimester mostly concerned retinoids for topical use (tretinoin and adapalene) (Fig 1; S4 Table). Exposure also decreased between the preconceptional period and the first trimester for methotrexate, topiramate, carbimazole, fluindione, atorvastatin, rosuvastatin and acetazolamide.

## Fetotoxic drugs

Regarding fetotoxic drugs, 155,514 (299 per 10,000) pregnancies were exposed during the second or third trimesters (Table 3). The most important exposure for fetotoxic drugs were NSAIDs for systemic use (128 per 10,000) and for topical use (122 per 10,000) (S5 Table). Ibuprofen was the most prescribed systemic NSAID and diclofenac was the most prescribed topical NSAID. We observed a decrease from 94,740 (182 per 10,000) exposed pregnancies during the second trimester, to 67,998 (132 per 10,000) during the third trimester (Fig 1; S5 Table). The decrease mostly concerned NSAIDs for topical use and NSAIDs for systemic use. Exposure to contraceptives increased from the second to the third trimester, mostly for etonogestrel and intrauterine devices with progestogens.

## Trends over the years from 2013 to 2019

We observed a decrease in the prevalence of the exposure from 2013 to 2019 for both teratogenic and fetotoxic drugs.

We observed a temporal decrease in the number of exposed pregnancies to teratogenic drugs from 8,898 (112 per 10,000) exposed pregnancies in 2013 to 5,234 (73 per 10,000) in

**Table 3. Maternal characteristics and pregnancy outcomes among pregnancies exposed to a fetotoxic drug.**

| | During T2 or T3 | During T2 | During T3 |
|---|---|---|---|
| Exposed pregnancies to a fetotoxic drug, n (% over all pregnancies)* | 155,514 (3.0%) | 94,740 (1.8%) | 67,998 (1.3%) |
| Exposed pregnant women | 151,128 | 92,838 | 66,734 |
| **Maternal age (years)** | | | |
| Mean (+/- SD) | 30.2 +/- 5.9 | 30.3 +/- 5.9 | 30 +/- 5.8 |
| < 20 | 4,878 (3.1%) | 2,776 (2.9%) | 2,396 (3.5%) |
| 20–29 | 66,556 (42.8%) | 40,070 (42.3%) | 29,272 (43.1%) |
| 30–39 | 74,882 (48.2%) | 45,784 (48.3%) | 32,687 (48.1%) |
| $\geq$ 40 | 9,198 (5.9%) | 6,110 (6.5%) | 3,643 (5.4%) |
| **Maternal comorbidities** | | | |
| Psychiatric troubles | 9,365 (6.0%) | 5,850 (6.1%) | 4,104 (6.0%) |
| Pre-gestational diabetes | 1,951 (1.3%) | 1,365 (1.4%) | 748 (1.1%) |
| Hypertension | 4,506 (2.9%) | 3,195 (3.4%) | 1,809 (2.7%) |
| **Number of hospitalisations in the year prior to pregnancy** | | | |
| Mean (+/- SD) | 0.4 +/- 1 | 0.5 +/- 1.1 | 0.4 +/- 0.9 |
| none, n (%) | 111,242 (71.5%) | 67,019 (70.7%) | 49,180 (72.3%) |
| 1 | 30,926 (19.9%) | 19,288 (20.4%) | 13,191 (19.4%) |
| 2 or more | 13,346 (8.6%) | 8,433 (8.9%) | 5,627 (8.3%) |
| Low-income status** | 13,162 (8.5%) | 8,517 (9.0%) | 5,565 (8.2%) |
| **Pregnancies outcome** | | | |
| Live births | 152,933 (98.3%) | 92,381 (97.5%) | 67,719 (99.6%) |
| Medical termination <22GW | 1,226 (0.8%) | 1,226 (1.3%) | (0.00%) |
| Medical termination$\geq$22GW | 507 (0.3%) | 452 (0.5%) | 62 (0.1%) |
| Still births | 848 (0.6%) | 681 (0.7%) | 217 (0.3%) |
| **Gestational age at birth (for live births only)** | | | |
| Mean (+/- SD) | 39 +/- 1.8 | 38.9 +/- 2 | 39.2 +/- 1.4 |
| Premature birth < 37GW | 9,156 (5.9%) | 6,991 (7.4%) | 2,541 (3.7%) |
| <28GW | 368 (0.2%) | 368 (0.4%) | (0.00%) |
| [28–31] GW | 762 (0.5%) | 691 (0.7%) | 84 (0.1%) |
| [32–36] GW | 8,026 (5.2%) | 5,932 (6.3%) | 2,457 (3.6%) |

* Whole cohort study includes n = 5,210,429 pregnancies at 2[nd] or 3[rd] trimester, n = 5,210,429 pregnancies at 2[nd] trimester and n = 5,149,745 pregnancies at 3[rd] trimester

** Low-income status was defined as affiliation to CMUc

**Abbreviations:** CMU (*couverture maladie universelle*), GW (gestational week), SD (standard deviation)

Data are shown as mean (+/- SD) or n (%)

2019 (Fig 2; S6 Table). The decrease concerned mostly the preconceptional period, from 7,419 (94 per 10,000) exposed pregnancies in 2013 to 4,427 (62 per 10,000) exposed pregnancies in 2019. Regarding exposure to retinoids for topical use, there was a decrease in exposed pregnancies from 4,300 (53 per 10,000) in 2013 to 2,432 (34 per 10,000) in 2019. Regarding retinoids for systemic use during preconceptional period or T1, the number of exposed pregnancies tended to decrease from 107 (1 per 10,000) in 2013 to 81 (1 per 10,000) in 2019. The most prescribed retinoid drug was isotretinoin. Among teratogenic AED exposure, we found a marked decrease specifically for valproic acid from 532 (7 per 10,000) exposed pregnancies in 2013 to 83 (1 per 10,000) exposed pregnancies in 2019. Divalproate exposure also decreased from 297 (4 per 10,000) exposed pregnancies in 2013 to 31 (0.4 per 10,000) exposed pregnancies in 2019 (Fig 2; S6 Table). Regarding exposure to HMG Co-A reductase inhibitors

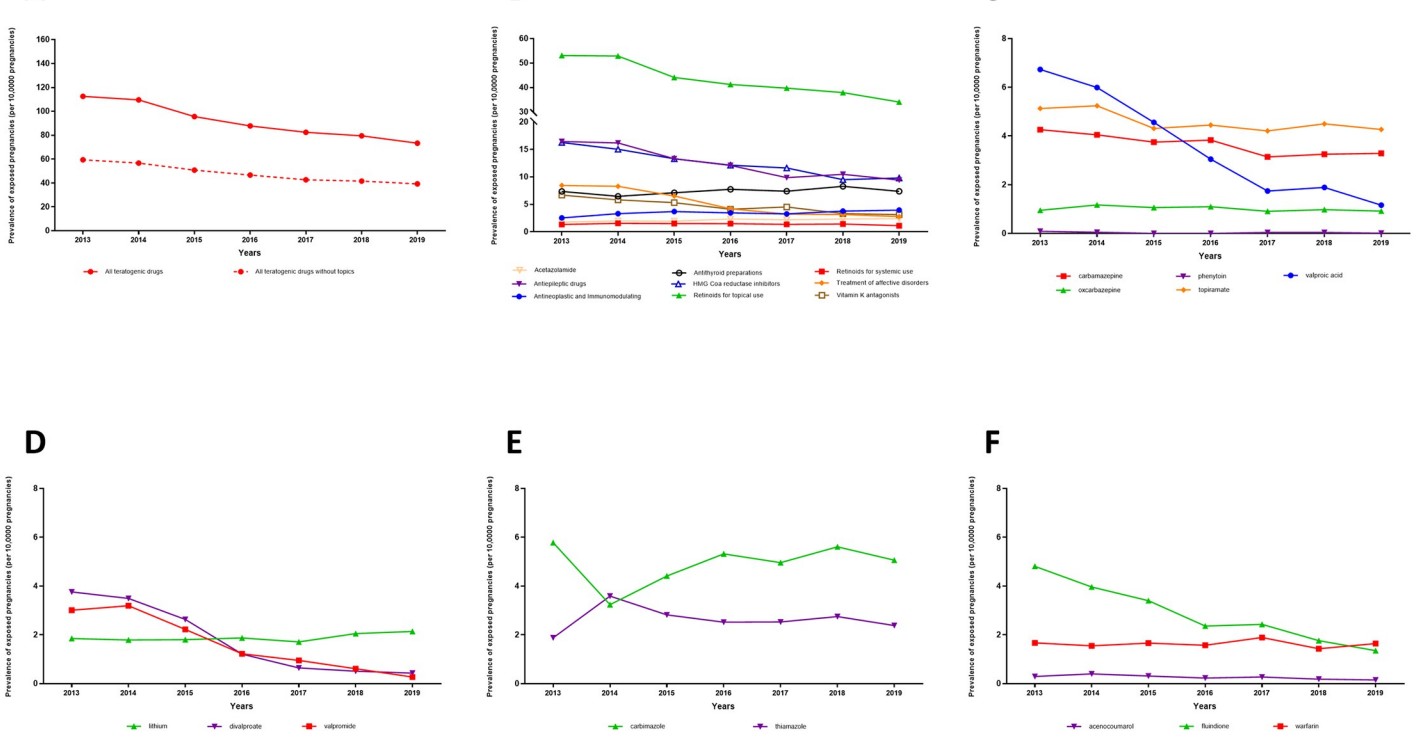

**Fig 2. Trends of exposure to teratogenic drugs during the preconceptional period or the first trimester for pregnancies starting between 2013 and 2019 in France.** The p-values indicate whether a statistically significant trend was observed between 2013 and 2019. **A.** Overall teratogenic drugs. Overall teratogenic drugs with topical retinoids and overall teratogenic drugs without topical retinoids. Trends were significant for: all teratogenic drugs ($p < 10^{-3}$), and all teratogenic drugs without topical forms ($p < 10^{-3}$). **B.** Overall teratogenic drugs. Trends were significant for: acetazolamide ($p < 0.05$), antiepileptic drugs ($p < 10^{-3}$), antineoplastic and immunomodulating ($p < 10^{-3}$), HMG Coa reductase inhibitors ($p < 10^{-3}$), treatment of affective disorders ($p < 10^{-3}$), vitamin K antagonists ($p < 10^{-3}$). **C.** Teratogenic antiepileptic drugs. Trends were significant for: carbamazepine ($p < 10^{-3}$), phenytoin ($p < 0.05$), topiramate ($p < 0.05$) and valproic acid ($p < 10^{-3}$). **D.** Drugs for affective disorders. Trends were significant for: divalproate ($p < 10^{-3}$) and valpromide ($p < 10^{-3}$). **E.** Antithyroid preparations. Trends were significant for: thiamazole ($p < 0.05$). **F.** Vitamin K antagonists. Trends were significant for: fluindione ($p < 10^{-3}$).

(statins), it decreased from 1,286 (16 per 10,000) exposed pregnancies in 2013 to 700 (10 per 10,000) exposed pregnancies in 2019. The most prescribed statin drug was atorvastatin.

We observed a decrease in the fetotoxic drug exposure from 27,188 (347 per 10,000) exposed pregnancies in 2013 to 15,928 (225 per 10,000) in 2019 (Fig 3, S7 Table). This decrease varied widely according to the type of drug. It was greatest among drugs with a higher prevalence of use during pregnancy such as topical NSAIDs, which decreased from 12,052 (154 per 10,000) exposed pregnancies in 2013 to 5,976 (84 per 10,000) exposed pregnancies in 2019. When looking at specific drugs, the decrease concerned mostly diclofenac for topical use from 8,680 (111 per 10,000) exposed pregnancies in 2013 to 4,270 (60 per 10,000) exposed pregnancies in 2019. Exposure to NSAIDs for systemic use also decreased from 11,266 (144 per 10,000) exposed pregnancies in 2013 to 6,810 (96 per 10,000) exposed pregnancies in 2019. In 2019, 12,786 (181 per 10,000) pregnancies were still exposed to NSAIDs, including all routes of administration, during the second or third trimester.

## Sensitivity analysis

When considering exposure only for drugs administered orally (excluding topical drugs), 128,085 (244 per 10,000) pregnancies were exposed to at least one harmful drug, corresponding to 25,671 (49 per 10,000) pregnancies with perinatal or first trimester exposure to a

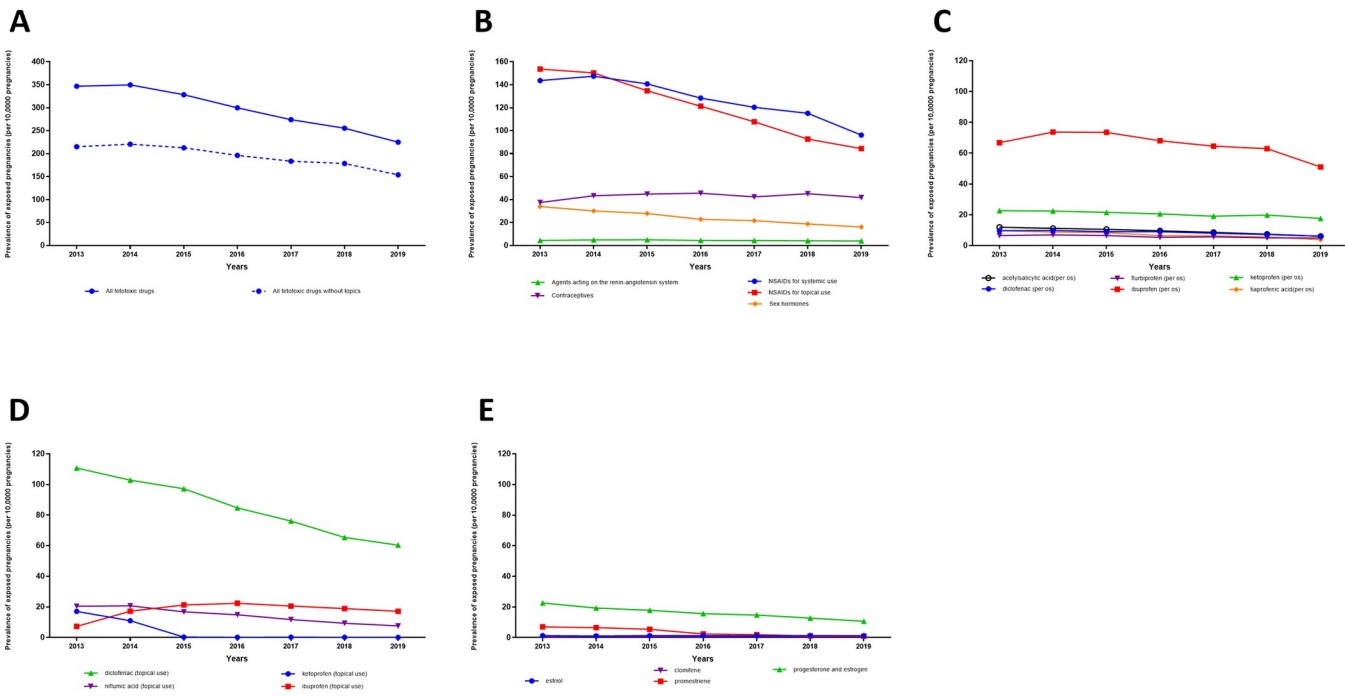

**Fig 3. Trends of exposure to fetotoxic drugs during the second or the third trimester for pregnancies starting between 2013 and 2019 in France.** The p-values indicate whether a statistically significant trend was observed between 2013 and 2019. **A.** Overall fetotoxic drugs. Overall fetotoxic drugs with topical NSAIDs and overall fetotoxic drugs without topical NSAIDs. Trends were significant for: all fetotoxic drugs ($p<10^{-3}$), and all fetotoxic drugs without topical forms ($p<10^{-3}$). **B.** Any fetotoxic drug. Trends were significant for: contraceptives ($p<10^{-3}$), non-steroids anti-inflammatory drugs for systemic use ($p<10^{-3}$), non-steroids anti-inflammatory drugs for topical use ($p<10^{-3}$), sex hormones ($p<10^{-3}$). **C.** NSAIDs for systemic use. Trends were significant for: acetylsalicylic acid ($p<10^{-3}$), diclofenac ($p<10^{-3}$), flurbiprofen ($p<0.05$), ibuprofen ($p<0.05$), ketoprofen ($p<0.05$), tiaprofenic acid ($p<10^{-3}$). **D.** NSAIDs for topical use. Trends were significant for: diclofenac ($p<10^{-3}$), ibuprofen ($p<10^{-3}$), ketoprofen ($p<10^{-3}$), niflumic acid ($p<10^{-3}$). **E.** Sex hormone. Trends were significant for: clomifene ($p<0.05$), progesterone and estrogen ($p<10^{-3}$), promestriene ($p<10^{-3}$).

teratogenic drug, and 101,619 (195 per 10,000) pregnancies with exposure to a fetotoxic drug during the second or the third trimester. Maternal characteristics and pregnancy outcomes for this sensitivity analysis were detailed in S8–S10 Tables (Supporting Information).

## Discussion

### Main findings

In this nationwide cohort study based on about 5 million pregnancies in France between 2013 and 2019, exposure to at least one harmful drug occurred in 389 per 10,000 pregnancies, with 92 per 10,000 pregnancies exposed to teratogenic drugs in the first trimester and 299 per 10,000 pregnancies exposed to fetotoxic drugs in the second or third trimester. Exposure to teratogenic slightly decreased over the study period. The main teratogenic drugs were retinoids, with topical forms accounting for about half of the exposures. The main fetotoxic drugs were NSAIDs, with systemic forms accounting for about 40% of the exposed pregnancies.

### Patterns of exposed pregnancies

The exposed pregnancies had a higher proportion of women with comorbidities, such as psychiatric disorders, pre-gestational diabetes, or hypertension, compared to the overall population. For instance, psychiatric disorders concerned 14.5% of pregnancies exposed to teratogenic drugs, whereas they concerned 3.9% of unexposed pregnancies. Similarly,

hospitalization during pregnancy was more frequent among exposed pregnancies. Economic status might also influence exposure to harmful drug during pregnancy, as 7.8% of women had low-income status in exposed pregnancies, *versus* 4.5% among unexposed pregnancies. A history of chronic disease and low-income status were previously reported to be related to harmful drug exposure during pregnancy in France [17].

## Changes in exposure during the course of pregnancy

Our study confirmed previous reports describing a decrease in harmful drug exposures during the course of pregnancy. In 2021, a study reported a decrease from the first to the third trimester, from 370 to 110 per 10,000 pregnancies and from 56 to 200 per 10,000 pregnancies using the Swedish and the Australian classification, respectively [17]. Our study complemented these data by describing drug exposures in relation to the specific critical periods of exposure, in order to account for fetal risks according to different susceptibilities to embryonic and fetal harm. Our study also provided comprehensive information by describing exposure to teratogenic drugs in the 90 days before the pregnancy, a period that might also harm the fetus if treatment is pursue during the first week of pregnancy. We observed a more than twofold decrease in teratogenic drug exposure between the preconception period and the first trimester, suggesting a drug discontinuation once pregnancy was diagnosed. However, this might be too late to avoid exposures during early embryonic development. For instance, the critical period for drugs inducing a neural tube defect is in early pregnancy up to six weeks of gestation [22]. To lower the prescription rates, patients and prescribers must be aware of the risks. They must be aware of the need for contraception in case of harmful treatment, especially for women with chronic disease. These findings support the importance of planning pregnancies for these patients and the role of preconceptional medical consultation. In addition, we observed a slight decrease for teratogenic antiepileptic drugs, treatment for affective disorders, and HMG Co-A reductase inhibitors from the second to third trimesters. This suggested the difficulty of stopping chronic treatment and sometimes the lack of alternative drugs.

## Teratogenic antiepileptic drugs

Valproic acid and derivatives (divalproate and valpromide) are proven teratogens, resulting in a risk of major congenital malformations, including the fetal valproate syndrome consisting in neural tube defects and congenital heart disease [23–26]. Foremost, prenatal exposure was also shown to be related to neurodevelopmental disorders such as autism and attention-deficit hyperactivity disorders [3–5]. In Europe, 51 per 10,000 pregnancies were exposed to AEDs in the 1999–2007 period [27], with valproic acid in monotherapy accounting for 23.7% of prescriptions [28]. However, since the worldwide recognition of the harmful effects of valproate derivatives in 2014, authorities endorsed measures to avoid exposure [29, 30]. The European Medicines Agency (EMA) first warned of the need for contraception during treatment in 2014, then contraindicated the prescription of valproate derivatives to women of childbearing age. Following these recommendations, as reported in a previous study in France [31], the use of older AEDs (such as valproic acid but also clonazepam and carbamazepine) decreased from 2007 to 2014 (-69.4%) whereas the use of newer AEDs with no suspected teratogenic effects (such as lamotrigine, pregabalin, levetiracetam, topiramate, gabapentin, and oxcarbazepine) increased (+73.4%). Our findings suggested that the EMA measures had a beneficial impact in France, since there was a sharp decrease in valproic derivatives prescriptions during the 2013–2017 period which persisted through 2019. In 2019, we thus reported that valproic acid exposure concerned 1 per 10,000 pregnancies and divalproate 0.4 per 10,000 pregnancies, which is much less than exposure in 2013 (with valproic acid in 7 per 10,000 pregnancies and

divalproate in 4 per 10,000 pregnancies). In Switzerland, a similar decrease was observed from 4 to 1 per 10 000 pregnancies in the 2015–2018 period [32]. These findings supported the effectiveness of strong recommendations and public health initiatives.

## NSAIDs

NSAID exposure in late pregnancy may induce irreversible fetal renal insufficiency, resulting in oligohydramnios, as well as premature closure of the arterial duct that can result in fetal demise [6–9]. Therefore, NSAIDs should not be used after 20 weeks' gestation [33, 34]. Differences were reported regarding systemic NSAID prescriptions in the world. For instance, a survey based on questionnaires from 18 countries assessed the use of anti-inflammatory and antirheumatic drugs (the M01 ATC class, including some systemic corticosteroids) during the 2011–2012 period. Exposure ranged from 220 per 10,000 pregnant women in Western Europe to 1,710 per 10,000 in South America with a decreasing trend from the second trimester (420 per 10,000 pregnant women) to the third trimester (360 per 10,000 pregnant women) [35]. Exposure was lower in Northern Europe, such as in Sweden and Norway, where it ranged from 30 per 10,000 pregnancies in the second trimester to 10 per 10,000 pregnancies in the third trimester during the 2005–2015 period [36, 37]. In France, during the 2011–2014 period, based on a national health insurance database random sample, exposure to anti-inflammatories and antirheumatic drugs (including systemic corticosteroids) was lower than in other Western European countries, ranging from 140 per 10,000 pregnancies in the second trimester to 50 per 10,000 pregnancies in the third trimester [14]. Considering systemic NSAIDs, exposure ranged from 120 per 10,000 pregnancies in the second trimester to 40 per 10,000 pregnancies in the third trimester. In our study, we observed a decrease in NSAIDs exposure between 2013 and 2019 of about one third. One could hypothesize that national French recommendations in 2005 and 2009 [38–40] to avoid NSAIDs during pregnancy had an impact, although it remained insufficient. In January 2017 the latest national warning to date from the French Medicine Agency was published. However, we did not observe a dramatic change in NSAID exposure during pregnancy after this date. Regarding topical use, the complications after maternal exposure are suspected based on case reports describing use of topical diclofenac [41, 42]. Our results were in accordance with another French cohort study, describing 60 per 10,000 exposed pregnancies after the sixth month [17]. This persistently high prevalence of exposure to topical NSAIDs during pregnancy in France contrasted with a two-fold lower in other countries over the 2011–2012 period [35].

## Retinoids

Retinoids are mainly used for treatment of severe acne and psoriasis. In case of exposure of the embryo, major birth defects have been described since 1985 [43] They were also recently associated with a possible risk of neurodevelopmental disorders [44]. Thus, retinoids are contraindicated in women planning a pregnancy and during pregnancy. Moreover, although the association between topical use and major congenital malformations has not been proven [45], the EMA contraindicated their prescription before and during pregnancy [44]. In our study, we identified a low exposure of systemic retinoids in accordance with previous studies from France and other countries [17, 36]. Given the teratogenic nature of retinoids, a Pregnancy Prevention Program (PPP) was developed to structure prescriptions since 1997 in France and 2003 in Europe. An evaluation of the PPP showed a decrease in the incidence of pregnancies beginning during treatment [46]. Nevertheless, the proportion of women receiving retinoids without any contraception remained constant (30%) [46, 47]. The prevalence of topical use during the preconceptional period or first trimester in our study (34 per 10,000 pregnancies in

2019) was in accordance with another national study and indicated that their prescription was more frequent in France than in other countries [17]. For instance, adapalene was prescribed in 3 per 10,000 pregnancies in New Zealand [48], *vs*. in 23 per 10,000 pregnancies in our study, and topical tretinoin in 6 per 10,000 in the Netherlands [27], *vs*. in 11 per 10,000 pregnancies in our study. Particular attention should focus on topical use, and its association with major congenital malformation should be further assessed.

### Limitations and strengths

First, the SNDS comprised reimbursement data but did not provide information on actual drug consumption. Our methodology could not quantify drug doses or duration of exposure. We also might have overestimated drug exposure if a drug had been purchased during the pregnancy and consumed after delivery. For instance, contraceptive exposure in the third trimester of pregnancy was very high, mostly for etonogestrel implants. We may suppose that women purchased the contraceptive during the pregnancy in anticipation for administration in the maternity ward after delivery. Similarly, NSAID prescriptions may have been filled during the third trimester in anticipation of use to relieve *postpartum* pain. Secondly, we had no information about drugs available over the counter (OTC) as they were not reimbursed. We might thus underestimate exposure related to OTC purchase, which include certain NSAIDs. However, underestimation was probably limited because most NSAID sales (70% of ibuprofen sales) were reimbursed in France [49, 50].

Our study had numerous strengths. The SNDS was a comprehensive nationwide French database covering 98.9% of the French population [21]. In our study, maternal and neonatal characteristics strongly concorded with previous published national data such as the National Perinatal Investigation [51]. To date, this was the first nationwide cohort study to analyse exposure to harmful drugs over a long period of time, as well as the first to distinguish relevant critical periods of exposure in pregnancy according to embryonic or fetal risks.

### Conclusion

This nationwide cohort study provided a comprehensive analysis of teratogenic and fetotoxic drug exposures during pregnancy, according to relevant critical periods. About one in 25 pregnancies was exposed to at least one harmful drug during a critical period, mainly NSAIDs (topical diclofenac and oral ibuprofen) and topical retinoids. The prevalence of teratogenic and fetotoxic drug exposure decreased significatively and reassuringly during the study period. However, the use of NSAIDs during the second and the third trimesters remained concerning. Our findings underline the importance of preconceptional medical consultation in order to optimize therapies with drugs compatible with pregnancy, especially in patients with chronic diseases.

### Supporting information

**S1 Table. Discharge codes or reimbursed drugs related to pregnancy outcomes and maternal conditions.**
(PDF)

**S2 Table. List of harmful drugs.**
(PDF)

**S3 Table. Maternal characteristics and pregnancy outcomes among pregnancies exposed to a teratogenic drug.** Preconceptional period, T1, T2 and T3.
(PDF)

**S4 Table. Teratogenic drug exposure according to pregnancy period.** Number of pregnancies (rate per 10,000 pregnancies).
(PDF)

**S5 Table. Foetotoxic drug exposure according to pregnancy period.** Number of pregnancies (rate per 10,000 pregnancies).
(PDF)

**S6 Table. Trends in teratogenic drug exposure during pregnancy between 2013 and 2019.** Number of pregnancies (rate per 10,000 pregnancies).
(PDF)

**S7 Table. Trends in foetotoxic drug exposure during pregnancy between 2013 and 2019.** Number of pregnancies (rate per 10,000 pregnancies).
(PDF)

**S8 Table. Maternal and pregnancy outcome characteristics (sensitivity analysis without topical retinoids and topical NSAIDs).**
(PDF)

**S9 Table. Maternal and pregnancy outcome characteristics among pregnancies exposed to at least one teratogenic drug (sensitivity analysis without topical retinoids).**
(PDF)

**S10 Table. Maternal and pregnancy outcome characteristics among pregnancies exposed to at least one fetotoxic drug (sensitivity analysis without topical NSAIDs).**
(PDF)

**S1 Fig. Flow chart.**
(PDF)

## Acknowledgments

We would like to thank Fédération Hospitalo-Universitaire Fighting Prematurity (FHU Prema) for its assistance.

## Author Contributions

**Conceptualization:** Margaux Louchet, Jeanne Sibiude, Laurent Chouchana.

**Data curation:** Mathis Collier, Nathanaël Beeker.

**Investigation:** Mathis Collier, Nathanaël Beeker.

**Methodology:** Margaux Louchet, Mathis Collier, Nathanaël Beeker, Jeanne Sibiude, Laurent Chouchana.

**Project administration:** Jean Marc Treluyer.

**Resources:** Jean Marc Treluyer.

**Supervision:** Jeanne Sibiude, Laurent Chouchana, Jean Marc Treluyer.

**Validation:** Jean Marc Treluyer.

**Writing – original draft:** Margaux Louchet.

**Writing – review & editing:** Laurent Mandelbrot, Jeanne Sibiude, Laurent Chouchana, Jean Marc Treluyer.

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
