## [Decision Letter · Decision Letter 0]

7 Jun 2023

PONE-D-22-23160Trends in harmful drug use during pregnancy in France between 2012 and 2018 : a nationwide cohort studyPLOS ONE

Dear Dr. Louchet,

Thank you for submitting your manuscript to PLOS ONE. After careful consideration, we feel that it has merit but does not fully meet PLOS ONE’s publication criteria as it currently stands. Therefore, we invite you to submit a revised version of the manuscript that addresses the points raised during the review process.

- Even, there is only one reviewer comments, the reviewer ask for several major changes in the manuscript. Please take time to answer (and modify according) each remarks.- Please add, recent years in your analysis if possible- Please provide a graphical abstract in eSupplement- Please be sure than all assumptions discussed in the approriate section are based on your own results

We look forward to receiving your revised manuscript.

Kind regards,

Jean Baptiste Lascarrou

Academic Editor

PLOS ONE

Journal Requirements:

Reviewers' comments:

Reviewer's Responses to Questions

**Comments to the Author**

1. Is the manuscript technically sound, and do the data support the conclusions?

Reviewer #1: Partly

2. Has the statistical analysis been performed appropriately and rigorously? 

Reviewer #1: Yes

3. Have the authors made all data underlying the findings in their manuscript fully available?

Reviewer #1: Yes

4. Is the manuscript presented in an intelligible fashion and written in standard English?

Reviewer #1: Yes

5. Review Comments to the Author

Reviewer #1: Trends in harmful drug use during pregnancy in France between 2012 and 2018: a nationwide cohort study

Peer Review Report

General comments:

This research has public health implications for France and for other countries that monitor the medicines used in pregnancy. Similar studies have been published recently indicating a high use of harmful medicines including teratogenic medicines. This study has the potential to complement the existing data on medicine use in pregnancy in France. The current study should also attempt to describe why the findings of this study are different from the other studies.

The study presented prevalence of medicine use in pregnancy, specifically in the three months pre-pregnancy and three trimesters of pregnancy. This is a very useful presentation of results to understand medicine use in pregnancy, and exposure of specific pregnancy related periods which have different susceptibility to foetal and embryonic harms. The medicines were categorised as teratogenic and fetotoxic, based on the pregnancy-related period when these medicines are deemed harmful. However, data was not presented for all the pregnancy-related time periods for fetotoxic medicines.

The study found that topical retinoids and systemic NSAIDs were the most used harmful medicines in pregnancy in France between 2012 and 2018. The authors note that topical retinoids are not considered as harmful as the systemic retinoids, but that these are contraindicated for use in pregnancy, by the regulatory authorities. I feel the study would benefit from a sensitivity analysis to demonstrate prevalence rates with and without retinoids in the mix.

In my opinion, the study results should be reported in a positive light, that other more harmful systemic medicines were rarely used in France in the time period considered. Also, a thorough spell check, language check and rephrasing of several paragraphs is recommended.

Introduction:

The section needs a thorough rewriting in light of the aims of the study. It is not clear what each paragraph aims to do. I have provided some big picture feedback below:

1. The following statements should be rephrased to describe what is intended from this paragraph:

“Most previous studies evaluate prevalence of potentially harmful drug using the FDA risk classification system. But since 2015, it has been abandoned and replaced by a narrative description of fetal risks of drugs (9).”

I think the authors want to describe the inconsistencies in the prevalence of medicine use in pregnancy, because of the different classification systems used. It is not clear why the authors point out that the FDA’s alphabetic classification system is no longer in use- what woud the authors like to imply from this sentence?

This paragraph needs further work- it specifically needs clarifying as to what the aim is: It seems like the authors intend to compare French utilisation rates with other developed countries, and describe the reasons for differences. However, this is not clear from the paragraph.

2. The final paragraph should summarise the gaps in the published French literature. This paragraph only points out one study that reported 2.4% of pregnancies exposed to NSAIDs. I believe other published French studies on medicines used in pregnancy have used some form of risk classification system. This paragraph or the one before should point this out clearly, and the resulting prevalence estimates as well, for the reader to make sense of what exists, and what gaps are being fulfilled by this study.

Methods

The study included pregnancies with date of conception between 2012 and Dec 2018, meaning that some pregnancies resulted in a birth in 2019 as well. Perhaps the authors should revisit how the data is presented. Because maternal characteristics and use is described during pregnancy, the data on utilisation (including graphs) should account for date of the birth instead of the date of conception, for a better representation of data. This is relevant because the prescriptions would have been filled in 2019, not 2018, for women who conceived in December 2018.

Results:

Table 1:

This table is very informative with details of the cohort and has implications for generalisability.

The abbreviations need explanatory footnotes.

In my opinion, Table S4 is more informative for the results section because it describes use by specific trimesters. Could the authors amend table 1 to reflect trimester-wise use? In the introductory paragraphs, the authors claim that no other French study provides data by trimester. This table might fulfill that gap- and it is relevant to fetotoxic medicines as well.

Figure 1:

Contraceptive use in T3 is very high. The discussion does not really go into the details of why some medicines or medicine groups were more prevalent in one trimester but not the other.

Secondly, regarding contraceptives, why did its prevalence not show in pre-pregnancy period? I would have expected to see high peaks of Contraceptives in pre-pregnancy and T1. In T1 I assume women receive a dispensing until they become aware of the pregnancy status. Likewise, for sex hormones (includes fertility medicines). I see that pre-pregnancy and T1 use was not described for Fetotoxic medicines which are contraindicated in later trimesters. Even so, it is good to present this data especially for international comparisons, given that these medicines are mostly categorised teratogenic using pregnancy risk categorisation systems.

Psychiatric disorders were found in 2% pregnancies, and 12% in women exposed to teratogenic meds.- how does that compare with antiepileptic use and psychiatric medicine use? I feel its worth commenting in the discussion section.

Discussion

1. I think the discussion needs more argument and placing the study in existing literature. Recent French studies have found much higher prevalence of teratogenic medicine use. What are the probable reasons that this study found such low rates? It is worth comparing these estimates with the present study. Having said that the discussion also needs a paragraph on what this study adds to existing data on medicine use in pregnancy in France. As the authors have pointed out, a few studies have reported use of harmful medicines in pregnancy in recent years. How does the current study complement those results?

2. While the authors discuss trends of use of broad medicine groups in the discussion on temporal trends through pregnancy and through the calendar years, specific medicines are not discussed. For example, the figures show Lithium use has increased- what are its implications?

Conclusions may be described differently-

Firstly, the results do not indicate decreasing exposures in sensitive periods (T1 presumably). However, the conclusions indicate so.

Secondly, the paper claims that the results indicate the importance of keeping healthcare professionals and patients informed. This aspect does not shine through in the methods and results. Were there ongoing interventions in France that aimed to increase the understanding of teratogenicity and fetotoxicity? If so, this should be addressed in the discussion.

6. PLOS authors have the option to publish the peer review history of their article (what does this mean?). If published, this will include your full peer review and any attached files.

Reviewer #1: **Yes: **Smriti Raichand

---

## [Author Response · Author response to Decision Letter 0]

30 Aug 2023

Dear Academic editor,

Plos One

Paris, July 20th 2023

Subject: Letter in response to points raised by the reviewer, Plos One.

Title: Trends in harmful drug use during pregnancy in France between 2012 and 2018: a nationwide cohort study

Article type: Original article, Nationwide cohort study

Dear Editor,

We are pleased to submit our revised manuscript entitled “Trends in harmful drug use during pregnancy in France between 2012 and 2018 : a nationwide cohort study” for publication in the Plos One.

We would like to thank the Editor for giving us the opportunity to provide a revised version of our manuscript and thank the reviewer for his/her comments. We believe these comments led us to improve the manuscript, including sensitivity analysis and an update regarding the year 2019. The tables for describing population cohort have also been updated with 2 new tables describing more precisely population exposed to teratogenic and fetotoxic drugs, respectively. We have to acknowledge that data regarding the pregnancies starting in early 2012 were not accurate regarding the preconceptional period occurring in 2011. To address this issue and ensure comparability between full calendar years, we choose to exclude 2012 from the study. We have updated the title of the manuscript in accordance.

We provide in a separate file a response to each point raised by the reviewer.

We hope that this revised version will fit with the Editor and Reviewer expectations for being suitable to be published in a high standard journal such as Plos One.

Your Sincerely,

Dr Margaux Louchet, MD

on behalf of all the author

---

## [Decision Letter · Decision Letter 1]

20 Sep 2023

PONE-D-22-23160R1Trends in harmful drug use during pregnancy in France between 2013 and 2019 : a nationwide cohort studyPLOS ONE

Dear Dr. Louchet,

Thank you for submitting your manuscript to PLOS ONE. After careful consideration, we feel that it has merit but does not fully meet PLOS ONE’s publication criteria as it currently stands. Therefore, we invite you to submit a revised version of the manuscript that addresses the points raised during the review process.

We look forward to receiving your revised manuscript.

Kind regards,

Jean Baptiste Lascarrou

Academic Editor

PLOS ONE

Reviewers' comments:

Reviewer's Responses to Questions

**Comments to the Author**

1. If the authors have adequately addressed your comments raised in a previous round of review and you feel that this manuscript is now acceptable for publication, you may indicate that here to bypass the “Comments to the Author” section, enter your conflict of interest statement in the “Confidential to Editor” section, and submit your "Accept" recommendation.

Reviewer #2: (No Response)

2. Is the manuscript technically sound, and do the data support the conclusions?

Reviewer #2: Partly

3. Has the statistical analysis been performed appropriately and rigorously? 

Reviewer #2: Yes

4. Have the authors made all data underlying the findings in their manuscript fully available?

Reviewer #2: Yes

5. Is the manuscript presented in an intelligible fashion and written in standard English?

Reviewer #2: No

6. Review Comments to the Author

Reviewer #2: Thanks for the resubmission attempt.

Overall comments:

- The authors have made changes to the manuscript but it still needs significant work

- The writing is not up to the standards of PlosOne. The language used is inconsistent and needs a lot of work.

Abstract:

- Methods: How long was the preconceptional period. An abstract is a standalone piece. Its more informative to a reader if you have described it up front, as the authors have done for trimester duration.

- Outcome: Drug exposure might need further description: Percents? No. of women exposed? No. of babies? Pregnancies?

- Results: Needs rewriting with better language, and careful use of tenses: For e.g. “One over 23 pregnant women”? Do the authors mean 1 in 23- i.e. about a quarter? ‘Exposure decreases’- perhaps change to ‘exposure to teratogenic and fetotoxic meds decreased from preconcpeiton through TO birth…. Something on these lines. Also, this is the first time the authors describe a ‘critical period’. This needs a mention in methods too. Could the authore ensure the difference between ‘use’ and ‘exposure’ from the iuyset? Because this is a database analysis, ‘use’ is not automatically presumed. Even so, if using ‘exposure’ please stick to that, or choose ‘use’ and describe in the main text that use may not be presumed but the term in the manuscript implies exposure.

- Conclusion: Language edits needed; unnecessary use of words like ‘providing a comprehensive analysis’- word space could be better utilised for further drawing conclusions

Introduction: The authors must aim to have a professional edit done for language and terminologies use. E.g. I am not sure if ‘sanitary disasters’ is a commonly used term for the thalidomide tragedy, and ‘dramatic injuries’ for foetal/congenital malformations? Also the wording can be more concise and succinct.

The authors claim the FDA classification was abandoned in 2015 (Line 70)- this needs revisiting. I am aware the FDA classification became descriptive, in place of the alphabetic (A,B,C,D,X)- this fact is not forthcoming. What do the authors mean by clinical terqatolofgist assessment in view of the classification? There are no references either?

Line 72: “In developed countries members of the OCDE, prescription of potentially harmful drug”….” What is OCDE? OECD? Please define at first use.

Because the author refer to FDA classification, perhaps describe what category X here is? Not all readers are aware how risk categories work?

Line 77-78: “Although prevalence of women who received a prescription of potentially harmful drug differed based upon the sources used…” What dothe authors mean by sources used- do they mean pregnancy risk classification systems?

Line 87: “Considering Non-Steroidal Anti-Inflammatory Drugs (NSAIDs), 2.3% of pregnancies were exposed after the sixth month of gestation….” – Is there a reason these medicines were highlighted in the introduction? Are these the most commonly used? I can see the point is using valproic acid and antiepileptic- even though it needs better argument here in the introduction- their potential to cause damage to foetus! The authors should highlight this to make their point, which will flow through to their results well later on. In line 90 the authors mention “This pharmacological class was the most prescribed among the harmful drugs.”- when? In the study reported in the previous line or in the current paper? If it’s the latter, this sentence must be deleted because the introduction should not report results of the paper. If it’s the former, the sentences need to be merged using better language and argument.

Line 91: “To date, no study has been performed to assess the exposure during pregnancy specifically according to the type of consequence for the fetus”- It is not clear what the authors are aiming for. What do they mean by ‘type of consequence’? Is that safety outcomes like foetal malformation?? Or clinical effectiveness?

Line 94: ‘critical period of exposure’ Again the authors do not describe what this means? This is the first time the term ahs been used- this should have been described in the prior paragraphs where international data was described. Also it is important to go into the detail of ‘critical’ period- what is it and why is it critical.

What was the rationale of presenting the trends of exposure in this study? This rationale needs better development.

METHODS:

- Line 103: “costly long term disease codes”- what do the authors mean? Does the SNDS include diagnosis codes of chronic diseases And how do they define ‘costly’- are these diseases costly to treat? Perhaps get rid of this term?

- Line 104-105: “corresponding to full reimbursement for low income”- does the data only have reimbursement for low income, or is that the only population for which the healthcare is reimbursed?

- It is not clear how the date of conception was estimated using the date of the last menstrual period or the gestational age- how many days/months prior? What was the calculation used: this is important because it will provide an insight into the estimated exposure timing

- Date of conception was used to define pregnancy: this should be stated upfront.

- Study provides exposure data on pregnancies between 2013 and 2020, as opposed to what authors claim: 2013 to 2019: here I would like to highlight ‘2020’ given that some pregnancies with conception date in Dec 2019 would end in a birth or would exceed the 22-week duration in the following year. Therefore, I hope the authors have cross checked the exposure dates in their datasets when they reported the results?

Spell check “Data analysis” subheading, check grammar and repetitions in Results and discussion throughout.

Results:

Given that the authors only report data for T1 and preconception for teratogenic meds, and for T2-T3 for fetotoxic meds, perhaps best to provide two separate graphs within the same figure. At the moment it looks like the cohort just never received NSAIDs for example. Also note that some info is only supplied in Supplementary tables: mainly the names of medicine groups and whether they are fetotoxic or teratogenic. Therefore, the reader has no way of knowing immediately whats happening in Figure 1. I quite like Figure 2 & 3 because it differentiates between teratogenic and fetotoxic meds.

A question- is it possible to provide a big picture graph for trends over the years for al harmful meds- wonder if that’s too many graphs- for fetotoxic and teratogenic together? You might choose to accept this comment or skip, but personally I feel it might give a neat overview of your findings.

In discussion section for antiepileptics- useful to remind the reader of what this study found before comparing to international data. In line 428 the authors say the valproate use decreased dramatically- how much was it?

The flow of discussion section needs work. It looks very hurried- perhaps the authors should spend more time finessing. An example: line 472: “Considering topical use, association with major congenital malformations among 215 exposed women has not been evidenced (47).” This sentence is incomplete and does not make sense.

Also please remind the reader of your findings in the discussion every time the current study is referenced so the reader may identify the magnitude of difference between the existing literature and current study without having to go through all the tables.

Having said that the arguments used in the discussion section have improved greatly, from the previous version. However, the language needs some marked changes.

Conclusions:

I feel the authors need to ‘sell’ this study better here- the study realy shows that the rates of systemic medicine use is comparable with the rest of the world, as opposed to what was initially imagined i.e. France has a higher use of harmful medicines. The authorities need to keep a tab on NSAIDs- true, but the authors need to provide options of what can be done for women who are in genuine need of these meds?

7. PLOS authors have the option to publish the peer review history of their article (what does this mean?). If published, this will include your full peer review and any attached files.

Reviewer #2: **Yes: **Smriti Raichand

---

## [Author Response · Author response to Decision Letter 1]

10 Oct 2023

Dear Academic editor,

Plos One

Paris, October 10th 2023

Subject: Letter in response to points raised by the reviewer, Plos One.

Title: Trends in harmful drug exposure during pregnancy in France between 2013 and 2019: a nationwide cohort study

Article type: Original article, Nationwide cohort study

Dear Editor,

We are pleased to submit our revised manuscript entitled “Trends in harmful drug exposure during pregnancy in France between 2013 and 2019 : a nationwide cohort study” for publication in the Plos One.

We would like to thank the Editor for giving us the opportunity to provide a revised version of our manuscript and thank the reviewer for his/her comments. We believe these comments led us to improve the manuscript.

We corrected the language in the manuscript by choosing the word “exposure” and defined it in the method section (i.e corresponding to drug dispensing). Additionally, to be in accordance with all manuscript, we changed “use” into “exposure” in the title.

In abstract and introduction, we defined more precisely the critical period. It referred to the period during which each drug is considered harmful: preconceptional period or first trimester for teratogenic drugs and second and third trimesters for fetotoxic drugs.

The tables for describing population cohort have been updated with one table describing more precisely pregnancies exposed to teratogenic drugs in Supporting Information.

We provide in a separate file a response to each point raised by the reviewer.

We hope that this revised version will fit with the Editor and Reviewer expectations for being suitable to be published in a high standard journal such as Plos One.

Your Sincerely,

Dr Margaux Louchet, MD

on behalf of all the author

---

## [Decision Letter · Decision Letter 2]

7 Nov 2023

PONE-D-22-23160R2Trends in harmful drug exposure during pregnancy in France between 2013 and 2019 : a nationwide cohort studyPLOS ONE

Dear Dr. Louchet,

Thank you for submitting your manuscript to PLOS ONE. After careful consideration, we feel that it has merit but does not fully meet PLOS ONE’s publication criteria as it currently stands. Therefore, we invite you to submit a revised version of the manuscript that addresses the points raised during the review process.

Dear Authors, Significant improvements has been made but it appears than your manuscript could benefit from editing by native english speaker. Some ressources are available on PLOS such as https://plos.org/resource/how-to-edit-your-work/ or Scribendi is able to perform it.Thanks in advance.

We look forward to receiving your revised manuscript.

Kind regards,

Jean Baptiste Lascarrou

Academic Editor

PLOS ONE

Journal Requirements:

Reviewers' comments:

Reviewer's Responses to Questions

**Comments to the Author**

1. If the authors have adequately addressed your comments raised in a previous round of review and you feel that this manuscript is now acceptable for publication, you may indicate that here to bypass the “Comments to the Author” section, enter your conflict of interest statement in the “Confidential to Editor” section, and submit your "Accept" recommendation.

Reviewer #2: All comments have been addressed

2. Is the manuscript technically sound, and do the data support the conclusions?

Reviewer #2: Yes

3. Has the statistical analysis been performed appropriately and rigorously? 

Reviewer #2: Yes

4. Have the authors made all data underlying the findings in their manuscript fully available?

Reviewer #2: No

5. Is the manuscript presented in an intelligible fashion and written in standard English?

Reviewer #2: No

6. Review Comments to the Author

Reviewer #2: Thanks for making the changes.

I feel the manuscript is in a much better condition than the previous versions.

Some of the errors here: It still needs some editing for language- keep tenses consistent- Suggest using past tense throughout. One over 25 means little- better to use one ‘in’ 25. Don’t capitalise drug names, stick to active voice throughout… etc.

Introduction:

• Can you give examples of manifestation of fetotoxity and teratogenicity in where you introduce the concepts?

• Do you know if the estimates of teratogenic medicine use in other countries included NSAIDs- perhaps say that somewhere?

• Break down the paragraph on aims of study. Don’t say ‘related’ critical periods: it doesn’t mean much- I am thinking related to what? A suggestion is to write …”….we assessesd exposure to… in the respective critical periods”….. or something to that effect.

•

Methods:

• While the text describes what a CMUc is, it still doesmt say if only low income indviduals were included in this study? Yes, I understand that CMUc provides free access to healthcare for low income and SNDS is a national health insurance. But its not clear if the database used consisted of low income as well as others? If yes, then did you study them separately? Would the data capture the drug use completely for all patients, or are there any differences in the data capture by population?

o The authors say that it is used as a proxy for economic status- meaning that anyone without a CMUc affiliation (registration? Status?) will be a ‘general population’ category as opposed to low income? This aspect really targets generalisability of this study.

• Could you supply ICD 10 codes used in this study? As a supplementary?

• Statistical analysis: please describe how you presented prevalence or exposure: as %? Although in some places it is n/10,000? Please harmonise this presentation.

Results:

• Because you describe CMUc in the methods, good to describe the breakdown of the cohort by CMUc category in the first paragraph of results too

• 6.2% premature births- can you please say what gestational age you consider premature in parantheses- I can see the table says it clearly to be 37w.

• Please also specify the prevalence in this paragraph along with the overall numbers- because you describe it in the table. Its easy to see the exposed pregnancies belonged to women with much more comorbidities, more CMUc candidates too.

• Subheading teratogenic drugs: can you add absolute no. of pregnancies before the prevalence per 10,000 in parentheses for AEDs and topical retinoids- just to give context and to harmonise presentation of results.

• Subheading fetotoxic drugs: the statement ….“We observed that exposure decreased from 94,740 pregnancies (181.8 per 10,000) during the second trimester, to 67,998 pregnancies (132.0 per 10,000) during the third trimester (Fig 1; S5 Table). Decrease mostly concerned NSAIDs for topical use and NSAIDs for systemic use”- while the absolute numbers decreased, the prevalence actually INCREASED from 181 to 132 per 10,000? You might want to revisit the calculations. I fear NSAID prevalence might have increased too? This might change your conclusions. Best to use the prevalence rates to describe decrease or increase and absolute numbers in parantheses to avoid confusion.

• It is clear the use of antiepileptics and statins were also very high. Can you comment on which ones in the paragraph on teratogenic meds? The information could be supplied here readily from the supplementary files and will add depth to your arguuments.

• Subheading on trends: In the statement “Regarding exposure to retinoids for systemic use, it decreased from 107 pregnancies (1.4 per 10,000) in 2013 to 81 exposed pregnancies during preconceptional period or T1”- do you mean to add a calendar year here?Also please add prevalence next to 81 pregnancies/\\.

• Can I ask the reason to separate valproic acid and divalproate? Its not common to see them differentiated. If they have similar effect on congenital defects then maybe keep them together and report the prevalence as one drug.

• Figure 2, subfigure “(B) Any teratogenic drug, excluding retinoids for topical use”- It is repeatition of fgure (A) just without topical retinoids. My suggestion in the first review was to provide a separate line trend for OVERALL teratogenic use with and without topical retinoids. I think a trendline is required for overall use here and in the next figure for fetotoxic drugs

• Also please stick to terms use or prevalence even for figures and tables;

• Thanks for adding the sensitivity analysis- this is great! But please stick to the same presentation for prevalence: Percentage or n/10,000.

• % with 2 decimal points is too much to read!

Discussion

• “Among maternal characteristics, women with comorbidities such as psychiatric disorders,354 diabetes or hypertension, were more likely to be exposed to a harmful drug during pregnancy.” I don’t think you have provided risk ratios here. Please avoid the use of term likely. Change to say cohort consisted of higher proportion of xyz compared to the pqrst cohort etc.

• “In our study, prescriptions for valproic derivatives dramatically decreased from 2013 to 2019, especially in the 2013-2017 period”- revisit phrasing of this sentence

• “since the valproate derivatives harmful effects mediatisation”- I am not sure if mediatasation is a word- what do you want to say- was it in the media a lot? Were there policy changes then? Please describe clearly.

• In the paragraph on AEDs while you describe all the literature, please make a comment on how your estimates relate to these international estimates- higher lower same or cant comment? Also no need to comment on age in the last sentence because tour study doesn’t consider it. How did the measures work in your case- your argument is that the measures came into effect in 2014-15 and there was decrease in use in Switzerland 2015-18. Your study was 2013 until 2019- what do you have to say about the impact off the EMA measures on use in France? This paragraph needs revisiting.

• NSAIDs: “Exposure is lower in Northern Europe such as in Sweden and Norway where it ranges from 0.3% in second trimester to 0.1% in third trimester (31,40). NSAIDs exposure is thus higher in France.”- nowhere before this sentence have you described the numbers in France. Perhaps where you say what your study found say what % or prevalence you found?

Strengths and limitations:

• Perhaps add a limitation that some pregnancy durations were estimated using gestational age assessment at birth- this could have caused some over- or under-estimation of exposure based on estimation of GA?

Conclusions:

Please make a comment regarding change in teratogenic and fetotoxic medicine use over years here- its great information!

7. PLOS authors have the option to publish the peer review history of their article (what does this mean?). If published, this will include your full peer review and any attached files.

Reviewer #2: **Yes: **Smriti Raichand

---

## [Author Response · Author response to Decision Letter 2]

26 Nov 2023

Dear Academic editor,

Plos One

Paris, November 26th 2023

Subject: Letter in response to points raised by the reviewer, Plos One.

Title: Trends in harmful drug exposure during pregnancy in France between 2013 and 2019: a nationwide cohort study

Article type: Original article, Nationwide cohort study

Dear Editor,

We are pleased to submit our revised manuscript entitled “Trends in harmful drug exposure during pregnancy in France between 2013 and 2019 : a nationwide cohort study” for publication in the Plos One.

We would like to thank the Editor for giving us the opportunity to provide a third revised version of our manuscript and thank the reviewer for his/her comments. 

We corrected the language in the manuscript by choosing prevalence (as number of exposed pregnancies per 10,000 identified pregnancies) to describe drug exposure and using the past tense in the all manuscript. We added ICD10 codes used to identify pregnancies. Moreover, we corrected the tables by using only one decimal point. 

Moreover, as recommended, a native English speaker reviewed the manuscript.

We provided a response to each point raised by the reviewer in a separate file.

We hope that this revised version will fit with the Editor and Reviewer expectations for being suitable to be published in a high standard journal such as Plos One.

Your Sincerely,

Dr Margaux Louchet, MD

on behalf of all the author

---

## [Editor Report · Decision Letter 3]

28 Nov 2023

PONE-D-22-23160R3Trends in harmful drug exposure during pregnancy in France between 2013 and 2019 : a nationwide cohort studyPLOS ONE

Dear Dr. Louchet,

Thank you for submitting your manuscript to PLOS ONE. After careful consideration, we feel that it has merit but does not fully meet PLOS ONE’s publication criteria as it currently stands. Therefore, we invite you to submit a revised version of the manuscript that addresses the points raised during the review process.

**Please correct a few grammatical errors and sentences difficult to read: **Introduction: lignes 68–70, Teratogenic drug exposure stopped shortly before conception is also of concern given the possible persistence of the drug in the body during early embryogenesis, perhaps before the mother becomes aware she is pregnant.Discussion: ligne 449, prescription was, not prescriptions was; ligne 458, or, not nor; lignes 463–464, Similarly, NSAID prescriptions may have been filled during the third trimester in anticipation of use to relieve postpartum pain; ligne 484: underline)

We look forward to receiving your revised manuscript.

Kind regards,

Jean Baptiste Lascarrou

Academic Editor

PLOS ONE
---

## [Author Response · Author response to Decision Letter 3]

29 Nov 2023

Dear Academic editor,

Plos One

Paris, November 29th 2023

Subject: Letter in response to points raised by the reviewer, Plos One.

Title: Trends in harmful drug exposure during pregnancy in France between 2013 and 2019: a nationwide cohort study

Article type: Original article, Nationwide cohort study

Dear Editor,

We are pleased to submit our revised manuscript entitled “Trends in harmful drug exposure during pregnancy in France between 2013 and 2019 : a nationwide cohort study” for publication in the Plos One.

We would like to thank the Editor for giving us the opportunity to correct the grammatical errors and sentences difficult to read.

We provided a response to each point raised by the reviewer in a separate file.

We also changed our financial disclosure by mentioning The Fighting Prematurity University Hospital Federation (FHU Prema) in Fundings.

We hope that this revised version will fit with the Editor and Reviewer expectations for being suitable to be published in a high standard journal such as Plos One.

Your Sincerely,

Dr Margaux Louchet, MD

on behalf of all the author

---

## [Editor Report · Decision Letter 4]

4 Dec 2023

Trends in harmful drug exposure during pregnancy in France between 2013 and 2019 : a nationwide cohort study

PONE-D-22-23160R4

Dear Dr. Louchet,

We’re pleased to inform you that your manuscript has been judged scientifically suitable for publication and will be formally accepted for publication once it meets all outstanding technical requirements.

Kind regards,

Jean Baptiste Lascarrou

Academic Editor

PLOS ONE
---

## [Editor Report · Acceptance letter]

19 Dec 2023

PONE-D-22-23160R4 

PLOS ONE

Dear Dr. Louchet, 

I'm pleased to inform you that your manuscript has been deemed suitable for publication in PLOS ONE. Congratulations! Your manuscript is now being handed over to our production team.

Kind regards, 

on behalf of

Dr. Jean Baptiste Lascarrou 

Academic Editor

PLOS ONE